# Occurrence and Distribution of Per- and Polyfluoroalkyl Substances from Multi-Industry Sources to Water, Sediments and Plants along Nairobi River Basin, Kenya

**DOI:** 10.3390/ijerph19158980

**Published:** 2022-07-23

**Authors:** Flora Chirikona, Natalia Quinete, Jesleen Gonzalez, Gershom Mutua, Selly Kimosop, Francis Orata

**Affiliations:** 1Department of Pure and Applied Chemistry, School of Natural Sciences, Masinde Muliro University of Science and Technology, P.O. Box 190, Kakamega 50100, Kenya; gmutua@mmust.ac.ke (G.M.); skimosop@mmust.ac.ke (S.K.); fomoto@mmust.ac.ke (F.O.); 2Department of Chemistry & Biochemistry, Freshwater Resources Division, Institute of Environment, Biscayne Bay Campus North Miami, Florida International University, 3000 NE 151st Street, North Miami, FL 33181, USA; nsoaresq@fiu.edu (N.Q.); jgonz1049@fiu.edu (J.G.)

**Keywords:** PFAS, distribution coefficient, Nairobi River, surface water, sediments, *Amaranthus viridis*

## Abstract

Per- and polyfluoroalkyl substances (PFAS) are ever-present pollutants in the environment. They are persistent and bio-accumulative with deleterious health effects on biota. This study assesses the levels of PFAS in environmental matrices along the Nairobi River, Kenya. An aggregate of 30 PFAS were determined in water, while 28 PFAS were detected in sediments and plants using solid phase extraction then liquid chromatography–mass spectrometric techniques. In water, higher levels of perfluoroundecanoic acids of up to 39.2 ng L^−1^ were observed. Sediment and plant samples obtained in the midstream and downstream contained higher levels of perfluorooctanoic acid of up to 39.62 and 29.33 ng g^−1^, respectively. Comparably, levels of long-chain PFAS were higher in water and sediments than in plants. Sediment/water log distribution of selected PFAS ranged between 2.5 (perfluoroundecanoic acid) and 4.9 (perfluorooctane sulfonate). The level of perfluorooctane sulfonate (1.83 ng L^−1^) in water is above the acceptable level in surface water posing high human health and ecological risks. The observed PFAS concentrations and distribution were attributed mainly to multi-industries located along the river, among other sources. The knowledge of PFAS occurrence and distribution in Nairobi River, Kenya, provides important information to local regulatory agencies for PFAS pollution control.

## 1. Introduction

Per- and polyfluoroalkyl substances (PFAS) are emerging man-made pollutants of great environmental concern. They are persistent, bio-accumulative, and toxic [1]. Some of the adverse health effects associated with exposure to PFAS include reduced kidney functioning, adverse pregnancy outcomes, thyroid disruption, and metabolic syndrome [2]. These compounds have cytotoxic and genotoxic potential for the human liver [3]. Some PFAS, such as perfluorooctane sulfonate (PFOS) as well as perfluorooctane sulfonyl fluoride (PFOSF) and related precursors, were added in May 2009 as the first fluorinated persistent organic pollutants (POPs) in the Stockholm Convention POPs list [4]. Furthermore, these chemicals are under consideration in the United Nation Environment Programme (UNEP) Strategic Approach to International Chemicals Management (SAICM) as a priority area. Perfluoroalkyl substances such as perfluorooctanoic acid (PFOA), its salts, and PFOA-related compounds were proposed for listing in the 2019 Convention on Persistent Organic Pollutants (COP) under the Stockholm convention [5]. 

Literature is available on the distribution of PFAS in numerous matrices such as municipal wastewater, surface water, rain water, sea water, ground water, soil, sediments, sewage sludge, and atmosphere, as well as in liver, serum, and tissue samples of human being and animals [6,7,8,9]. The majority of these studies have been undertaken in developed countries. There is limited information on the environmental distribution and emission of PFAS in Africa [10]. Assessment of PFAS discharge, pollution, and human exposure in developing countries is crucial due to the ever-growing population which is adopting new and sophisticated consumption habits [10]. According to the National Implementation Plan (NIP) of Kenya (2014), it was recommended that the levels of PFAS in different environmental matrices should be determined. In Kenya, PFAS have been previously reported in the River Sosiani, wastewater treatment plants within the Lake Victoria region and in Lake Victoria Gulf water [11,12,13]. The level of PFOA in River Sosiani ranged between 1.6 and 8.8 ng L^−1^. The main source of PFAS contamination in River Sosiani was industrial and domestic wastewater discharge [11]. The level PFOA and PFOS in wastewater treatment plants within the Lake Victoria region ranged between 1.3 and 28 ng L^−1^ and 0.9–9.8 ng L^−1^, respectively. The sources of PFAS in wastewater treatment plants were domestic, hospital, and industrial discharges [12]. The concentration range of PFOA and PFOS in Lake Victoria Gulf water was 0.4–11.7 and 0.4–2.53 ng L^−1^, respectively. The main source of PFAS contamination was industrial and domestic wastewater discharges [13]. There is a lot of literature on pollution of Nairobi River by different contaminants. However, none of these studies have reported on the levels of PFAS. We therefore focus on assessing the occurrence and distribution of PFAS in surface water, sediments, and plants from Nairobi River, Kenya. The water/sediments log distribution coefficient of selected PFAS along Nairobi River matrices was also determined.

## 2. Materials and Methods

### 2.1. Description of the Study Area

Eight sampling points were selected along the Nairobi River. The river cuts across Nairobi city which has a population of 4.379 million (2019 census report). The selected sampling points are located near numerous multi-industries such as cottage industries, among other potential PFAS sources. They included: Chiromo Bridge/Museum Hill roundabout, S1 (−1.274697, 36.811925), John Michuki Park, S2 (−1.275753, 36.817161), Globe Cinema roundabout, S3 (−1.278397, 36.820902), Kirinyaga road, S4 (−1.279823, 36.825661), Kariokor, S5 (−1.281559, 36.832641), Gikomba, S6 (−1.286105, 36.836928), Juja Outering road, S7 (−1.264953, 36.879202), Eastern bypass, S8 (−1.245381, 36.988022). Figure 1 shows the study area and the selected sampling points in the Nairobi River, Kenya.

These sampling points are mainly surrounded by low-income residential areas. The area is densely populated by residents who provide labor to the industries. Nairobi River is exposed to pollution from domestic, industrial, and agricultural discharges [14]. The descriptions of the major anthropogenic activities in the study sampling points are shown in Table 1. 

Water from the river is mainly used by the low-income residents for drinking, domestic purposes, and watering crops [14].

### 2.2. Chemicals and Materials 

HPLC-grade hexane, methanol, acetone, water, ammonium hydroxide, ammonium formate, and methylene chloride were acquired from Merck, through Kobian Kenya Ltd. An isotopically mass-labeled standard mixture containing 24 PFAS (MPFAC-24ES 1 μg mL^−1^ in methanol), a native standard mixture containing 30 PFAS (PFAC30PAR, 1 μgmL^−1^ in methanol), and a HFPO-DA-labeled standard solution (50 μg mL^−1^ in methanol) were acquired from Wellington Laboratories Inc. (Guelph, Ontario, Canada). Methanolic stock solution at a concentration of 1 μg L^−1^ was prepared and stored at −8 °C. To prepare working solution, water was used to dilute the stock solution to 1 and 10 ng mL^−1^. The diluted solutions were kept at 4 °C. A secondary standard used for initial calibration verification (ICV) contained 24 PFAS (PFC-24, 2 μg mL^−1^ in methanol:water (80:20), and it was acquired from AccuStandard (New Haven, CT, USA). Appendix A presents a comprehensive list of the PFAS in each internal and native standard. Solid Phase Extraction (SPE) cartridges (Oasis WAX 6CC VAC cartridges—150 mg) were purchased from Waters Corporation. Graphitized carbon black and Florisil were acquired from Fisher Scientific (Hampton, NH, USA). 

### 2.3. Sample Collection 

Samples were collected along Nairobi River in areas where detection of PFAS was most expected due to anthropogenic input. The sampling sites selected were near cottage industries, hospitals dumpsites, and residential areas. During sample collection, fluoropolymer-made containers were avoided to minimize PFAS contamination into the sample. All apparatus used during sample extraction and preparation were rinsed with methanol. Before collecting the surface water, the polypropylene bottles were rinsed thoroughly using the river water. The water samples were collected in triplicate using a pole dipper. The water samples were mixed before extraction to obtain a representative sample. Sediments samples and plant samples (*Amaranthus viridis*) were collected within a 1 m^2^ area. The collected samples were transported in ice boxes to Masinde Muliro University of Science and Technology (MMUST) Laboratory. The water samples were refrigerated at 4 °C before extraction. The sediments and plant samples were air dried to a constant mass. Five grams of the dried sediments and plant samples were packed in polypropylene paper and then transported to Florida International University (FIU) for extraction and analysis.

### 2.4. Extraction of PFAS from Water Sample 

The 500 mL water samples were filtered using Whatman No.1 filters papers to remove particulate matter. The particulate-free water samples were extracted using Oasis WAX SPE cartridges as described in the literature [15]. In the typical procedure, the cartridges were pre-conditioned with 4 mL of 0.1% methanolic NH_4_OH and 4 mL LC-MS grade water in that order. The samples were loaded under a vacuum at a rate of 3 mL per minute. The loaded cartridges were eluted with 4 mL MeOH. The eluent was concentrated to dryness and then reconstituted to 1 mL using MeOH. The extracts were transported in polypropylene vials to FIU for instrumental analysis. Before analysis, the extracted water samples were filtered using a polypropylene syringe with a glass fiber filter 0.45 µm.

### 2.5. Extraction of PFAS from Sediments and Plants 

Extraction of PFAS from sediment and plant samples was completed according to Lemos et al. [16], with slight modification. The dried sediment and plant samples were ground and homogenized using a pestle and a mortar. Then, 0.5 g of the homogenized sample was placed in a polypropylene centrifuge tube then spiked with 100 µL of 2.5 µg L^−1^ methanolic internal standard solution. Thereafter, 5 mL of methanol was added and the mixture was vortex mixed, sonicated for 15 min, and finally centrifuged for 10 min at 2000 rpm. The supernatant was decanted into another polypropylene tube. The extraction procedure was repeated using another 5 mL of methanol and both supernatant fractions were combined. Dispersive SPE (d-SPE) was used to clean the combined extract; whereas 50 mg florisil and 100 mg graphitized carbon black (GCB) were used for cleaning the sediment samples while 50 mg florisil and 200 mg GCB were used for cleaning the plant samples. After d-SPE, samples were vortex mixed and then centrifuged for 10 min at a speed of 10,000 rpm and the supernatant which contained the clearer extract was filtered using a syringe filter (acrodisc filter, pore size of 0.2 µm made with a propylene housing, glass fiber pre-filter, and polyethersulfone membrane) and transferred into an LC polypropylene vial for LC-MS/MS analysis.

### 2.6. Instrumental Analysis

Instrumental analysis was completed according to Li et al. [17]. PFAS analyses were conducted using an Agilent 1290 Infinity II liquid chromatograph coupled with an Agilent 6470 triple quadrupole (LC-MS/MS) equipped with a jet stream electrospray ionization source. PFAS free tubing and a delay column (Hypersil GOLD aQ C18, 20 × 2.1 mm, 12 μm) placed between the mobile phase mixer and the sample injector were used in the LC to avoid possible contamination. A Hypersil GOLD pentafluorophenyl (PFP) column (150 × 2.1 mm, 3 μm) fitted with a PFP guard column (Hypersil Gold PFP 5 μm drop-in guards) was used to separate PFAS analogues. The temperature and the flow rate were set at 50 °C and 0.4 mL min^−1^, respectively. The mobile phase used was methanol and 5 mM ammonium formate in LC-MS water and injection volume was 100 µL for water sample extracts and 20 µL for plant and sediment extracts. Appendix A show the LC gradient conditions, and the MS parameters, respectively. A dynamic multiple-reaction monitoring (MRM) technique in negative mode was used for sample acquisition and determination of multiple PFAS. Appendix A shows in detail the MRM method, which includes data on precursor and product ions that were monitored, their retention time, fragmentor voltage, and collision energy.

### 2.7. Quality Control

The methods used in this study were previously validated at FIU [16,17]. Procedure blank, spiked blanks, and matrix spiked samples were prepared and analyzed along each batch of samples. Dilutions from 30 PFAS native standards in concentration range of 2–1000 ng L^−1^ (for water samples) and 5–1000 ng L^−1^ (for plant and sediment samples) were used to prepare a 10-point calibration curve. Analytical curves were run in the beginning and end of every set of samples, together with an initial calibration verification (ICV) from a secondary standard solution at a concentration of 100 ng L^−1^ and a continuing calibration verification (CCV) after 7–10 samples. The CCV and ICV measured concentrations should not deviate more than 30% from the assigned value, otherwise appropriate corrections (cleaning and recalibration of the instrument) were made before proceeding with the following injections. Samples were diluted and re-run if sample concentrations were above the calibration curve range.

The calibration curves were plotted using the area ratio (peak area of compound/peak area of respective isotopically labeled IS) against the concentration of PFAS. Most of the compounds had linear curves with R^2^ coefficients greater than 0.99 at the studied concentration range. The method detection limit (MDL) for water was determined in a previous study [17] and ranged from 0.001 to 1.02 ng L^−1^ while for sediment and plant samples the detection was estimated from the lowest point of the chromatograms that produced a signal-to-noise of at least three for each compound ranging from 0.1 to 1 ng g^−1^ as shown in the Appendix A. 

### 2.8. PFAS Data Analysis and Statistics

Data analysis was performed using Mass Hunter QQQ Quantitative analysis software for peak integration and quantification. The following criteria had to be met for PFAS to be considered present and quantifiable: appearance of a peak within 0.2 min of the same retention time (RT) of corresponding isotopically labeled IS, existence of a confirmation peak when available and a PFAS signal to noise ratio (S/N) above 3. In the case of native standards without mass-labeled analog, the selection of the IS to be used was based on similarity in functional groups, chain length, and retention time as shown in Appendix A. Excel and Origin 2022 were used for statistical and graphical analysis.

## 3. Results and Discussion

A total of 30 PFAS were found in Nairobi River water, while 28 PFAS congeners were detected in sediments and plants. The list of PFAS abbreviations can be found in Appendix A. The total PFAS concentration in a sampling site was determined by the summation of the concentration of all PFAS congeners in the same location. Figure 2 shows the spatial distribution of PFAS in the Nairobi River water.

The total sum of PFAS (∑PFAS) in the river water decreased in the order: S4 Kirinyaga road (26.17 ng L^−1^) > S6 Gikomba (21.67 ng L^−1^) > S1 Chiromo bridge (19.17 ng L^−1^) > S5 Kariokor (18.94 ng L^−1^) > S2 John Michuki Park (17.90 ng L^−1^) > S7 Juja outering (10.15 ng L^−1^) > S3 Globle Cinema (7.88 ng L^−1^) > S8 Eastern bypass (6.70 ng L^−1^). The observed order clearly reflects the effect of proximity to numerous cottage industries and the location of a sampling point along the river. The upper stream and midstream sampling points, S4 Kirinyaga, S6 Gikomba, S1 Chiromo, S5 Kariako, and S2 John Michuki are located near numerous cottage industries such as paints, panel beating, car repair/garage, dyeing industries, and food vending industries and presented the higher PFAS levels. Industrial discharge is known to contribute heavily to the spread of PFAS in surface water [8]. Cottage industries are expected to be the major sources of the numerous PFAS in the Nairobi River. Downstream sampling points S7 and S8 are located near learning institutions with sparsely populated residential areas. These regions have very few cottage industries hence lower PFAS levels, whereas the distribution pattern of PFAS in the Nairobi River reflects land use patterns with higher PFAS levels usually occurring in industrialized and urbanized areas [18]. The sampling point S3 Goble cinema, though located upstream and near many cottage industries, had a relatively lower concentration of PFAS in water, which could be mainly attributed to the higher rate of PFAS adsorption in sediments; confirmed by the elevated PFAS levels in sediments from this region.

The level of PFAS in sediments from the Nairobi River Basin is higher compared to their corresponding PFAS in water. PFAS can strongly bind to organic matter in sediments via hydrophobic interactions, which makes sediments an important sink and reservoir of PFAS [19,20]. Spatial PFAS distribution in sediments is shown in Figure 3. 

In sediments, the level of ∑PFAS decreased in the following order: S7 Juja outering (51.93 ng g^−1^) > S6 Gikomba (48.47 ng g^−1^) > S3 Goble cinema (38.20 ng g^−1^) > S4 Kirinyaga (35.53 ngg^−1^) > S5 Kariokor (30.04 ng g^−1^) > S8 eastern bypass (26.27 ng g^−1^) > S1 Chiromo bridge (25.70 ng g^−1^) > S2 John Michuki (22.31 ng g^−1^). Midstream and downstream sampling points had higher levels of PFAS compared to upstream. This could be attributed to mass transportation and deposition of PFAS in sediments downstream and midstream. PFAS partition to sediment can be very complex and influenced by total organic carbon, nitrogen, and phosphorous contents, PFAS functional group [21], type and concentration of competing ions, and pH [8]. The pH could change PFAS speciation and sediment chemistry; therefore, affecting surface complexation and electrostatic processes. Inorganic cations can form ionic interactions between anionic charged heads of PFAS, resulting in higher PFAS adsorption into particulates [22,23].

Plant samples were collected in only six sites: S2, S3, S4, S5, S7, and S8. The PFAS levels obtained in plant samples were slightly lower (about 1.5 times lower) than in sediments. Figure 4 shows the spatial distribution of PFAS in plants in the studied area.

In plants, the level of PFAS decreased in the following order: S2 John Michuki 35.0 ng g^−1^ > S7 Juja Outering (23.76 ng g^−1^) > S8 Eastern bypass (22.88 ng g^−1^) > S5 Kariokor (17.96 ng g^−1^) > S3 Goble Cinema (16.52 ng g^−1^) > S4 Kirinyaga road (13.45 ng g^−1^). The primary pathway for translocation of PFAS from the environment (contaminated water and sediments at the riverbank) to plants is through the roots. The movement of PFAS from the roots to shoot is by both active and passive transport mechanisms. PFAS get accumulated in the cell wall, cell organelles, and the intercellular space within the cortex [24]. The different PFAS levels in plants could be influenced by abiotic factors, such as soil organic matter, pH, salinity, and temperature, which can affect PFAS adsorption [25]. Uptake of PFAS by plants is influenced by the compound’s physicochemical properties such as the chain length of the perfluorocarbon, its functional group, as well as its ability to dissolve in water and to evaporate [25].

The detection frequency of PFAS in water and sediment samples was higher than in plant samples. For instance, in water, PFBA, PFPeA, PFBS, 4-2 FTS, PFHxA, GenX, PFHpA, PBSA, 6-2 FTS, PFOA, PFNA, 8-2 FTS, PFDA, PFDS, PFUdA, PFDOA, and 8-2 FTS had the detection frequency of 100%. FHxSA and N-MeFOSAA (87.5%), PFHpS, PFONS and PFTeDA (75%), PFHxS (62.5%), and Adona (50%) were also frequently detected in surface water. Only four PFAS had a detection frequency below 50% in surface water; PFOUDS (37.5%), NEtFOSAA, (37.5%), PFPeS (25%), and PFNS (12.5%). In sediments, PFBA, PFPeA, PFBS, PFHxA, PFHpA, PFOA, PFOS, PFNA, FOSA, PFDA, PFUdA, FHxSA, PFDoA, N.MeFOSAA, N-EtFOSAA, PFTrDA, PFTeDA, and FOSA had 100% detection frequency. PFHxS and PBSA (85.5%), Adona and PFNS (75%), and PFONS (50%) were also frequently detected in sediments. Four PFAS had a detection frequency below 50% in sediments, PFOUDS (25.5%), 6-2 FTS (12.5%), PFHpS (12.5%), PFDS (12.5%), and 4-FTS was not detected above the MDL. A high detection frequency of PFAS in water and sediments indicates widespread occurrence, and this is of great concern since it can correlate to adverse effects on human health and ecology [6]. In plant samples, only five PFAS had a detection frequency of 100%; PFPeA, PFHpA, PFOA, PFOS, and PFDoA. The majority of the PFAS had a detection frequency below 50%, the detection frequency of PFHxS, Adona, and PFNA was 33.33%, the detection frequency of 4-2 FTS, 6-2 FTS, FBSA, PFNS, 8-2 FTS, PFDS, PFHxA, N-EtFOSAA, and FOSA was 16.67%, while PFHpS and PFOUDs were not detected above the MDL. Appendix A show the range, mean, total concentration, and the detection frequency of all PFAS detected in the Nairobi River water, sediments, and plants, respectively. The high detection frequency of PFAS in environmental matrices shows widespread occurrence and distribution of PFAS in the river basin.

PFAS can be classified as follows: perfluoroalkyl carboxylic acids (PFCA), perfluoroalkyl sulfonic acid (PFSA), perfluoroalkane sulfonamide (PFOSA), perfluoroalkane sulfonamido acetic acid (FOSAA), fluorotelomer sulfonic acid (FTS), and perfluoroether carboxylic acid (PFECA). When categorized in classes, by adding the concentration of all compounds of the same category and determining their percentage composition, the composition varied in the different sample types as shown in Figure 5.

The sum of PFCA and PFSA represents 70–90% of the total PFAS detected in river water, sediments, and plants. In water, PFCA (70.6%) was the most predominant class followed by FTS (23.5%), PFSA (3.9%), FOSAA (1.10%), PFOSA (0.479%), and then PFECA (0.425%). Among the PFCA in water, PFUdA (39.219 ng L^−1^) was the most predominant followed by PFBA (13.5 ng L^−1^), PFPeA (9.07 ng L^−1^), PFDoA (6.06 ng L^−1^), PFHxA (5.92 ng L^−1^), PFOA (5.9 ng L^−1^), PFHpA (4.49 ng L^−1^), PFNA (2.68 ng L^−1^), PFTeDA (2.17 ng L^−1^), and PFDA (1.50 ng L^−1^). This observation is contrary to previous studies around the world where PFOA was reported to be the most dominant compound among PFCA [26,27]. Among the FTS class, 6-2 FTS (29.3 ng L^−1^) had the highest concentration, while 4-2 FTS and 8-2 FTS had concentration levels below 1 ng L^−1^. Among the PFSA, PFOS (1.83 ng L^−1^) was the most predominant followed by PFDS (1.7 ng L^−1^) and PFBS (1.2 ng L^−1^). The concentration levels of PFPeS, PFHxS, and PFNS were below 1 ng L^−1^. The concentration of PFOS (1.83 ng L^−1^) was higher than the acceptable level in natural water. The European Union (E.U) has set environmental water quality standards (EWQS) that PFOS in inland natural surface water should be <0.65 ng L^−1^ and in the biota should be <9.1 ng g^−1^ w.w [28]. This is of great concern because exposure to PFOS and PFOA through water and food causes adverse health effects, for instance, in vitro and in vivo studies have shown that exposure to PFOS and PFOA causes damage of the liver, heart, lungs, and kidneys. It also alters normal activities of the nervous system, has adverse effects on both the local and systemic immune system. It affects both female and male fertility as well as development of offspring [29,30,31]. The levels of PFECA, FOSAA, and PFOSA in water were very low as shown in Figure 5.

In sediments, PFCA (53.2%) was the most predominant class, followed by PFSA (37.3%), FOSAA (5.47%), PFOSA (1.84%), FTS (1.72%), and PFECA (0.543%). Among the PFCA class, PFOA (39.62 ng g^−1^) was the most predominant, followed by PFPeA (37.98 ng g^−1^), PFBA (22.82 ng g^−1^), PFDA (11.42 ng g^−1^), PFHxA (10.6 ng g^−1^), PFUDA (8.0 ng g^−1^), PFHpA (6.5 ng g^−1^), PFDoA (6.3 ng g^−1^), PFTeDA (3.17 ng g^−1^), and PFNA (3.16 ng g^−1^). Among the PFSA class distribution in sediments, PFOS (41.16 ng g^−1^) was the most predominant followed by PFNS (31.03 ng g^−1^), PFBS (16.0 ng g^−1^), PFHxS (10.56 ng g^−1^), PFDS (1.56 ng g^−1^), and PFHpS (1.69 ng g^−1^). Among the FOSAA, N-MeFOSAA (8.0 ng g^−1^) was the most predominant followed by N-EtFOSAA (6.94 ng g^−1^). The levels of FTS, PFECA, and PFOSAA in sediments were very low as seen in Figure 5.

In plants, PFCA (79.8%) was also the most predominant followed by PFSA (12.4%), FTS (4.42%), FOSAA (1.24%), PFOSA (0.989%), and PFECA (0.543%) as seen in Figure 5. Among the PFCA, the most predominant was PFPeA (32.46 ng g^−1^) followed by PFOA (29.33 ng g^−1^), PFBA (28.23 ng g^−1^), PFHxA (5.21 ng g^−1^), PFHpA (3.67 ng g^−1^), PFDA (2.48 ng g^−1^), PFUDA (2.25 ng g^−1^), PFDoA (1.22 ng g^−1^), and PFTeDA (0.71 ng g^−1^). Among the PFSA, PFOS (5.68 ng g^−1^) was the most predominant followed by PFPeS (5.07 ng g^−1^) and PFHxS (3.78 ng g^−1^). The other PFSA such as PFBS, PFNS, and PFDS had levels below 1 ng g^−1^ while PFHpS was not detected. Among the FTS, the most predominant was 6-2 FTS (2.90 ng g^−1^) followed by 8-2 FTS (1.45 ng g^−1^) and 4-2 FTS (0.80 ng tg^−1^). The levels of FOSAA, PFOSA, and PFECA were generally lower in plant samples compared to PFCA, PFSA, and FTS as shown in Figure 5. The relatively low levels of FOSAA, PFOSA, and PFECA could be attributed to their low use in consumer products. Studies show that many PFAS precursors when released in the natural environment can undergo biotic and abiotic transformation, even if limited, generating perfluoroalkyl acids (PFAAs) over time [32].

When PFAS are categorized according to chain length, perfluoroalkyl substances with C4–C7 are regarded as short chain while those with C8 and above are regarded as long-chain PFAS [20]. Appendix A defines the classes according to chain length. Figure 6 shows PFAS distribution in water, sediments, and plants according to their chain length classification.

From the results, the sum of long-chain PFAS in the water and sediments was higher than short chain. In water, long-chain PFAS accounted for 72% while short-chain PFAS accounted for 28%. In sediments, long-chain PFAS accounted for 62% and short chain accounted for 38%. This is in commensurate with a previous report [33], which found higher levels of long-chain PFAS in sediments, evidencing their ability to bind strongly to particles. Chain length and functional group type are the main factors influencing PFAS distribution in environmental samples [34]. The low concentration of short-chain PFAS in sediments can be attributed to their higher solubility, which confers them more mobility and to be less adsorbed onto the sediments. The solubility of PFAS decreases with increased carbon chain length [34]. In plants, the sum of short chain accounted for 62%. Studies have shown that long-chain PFAS are less likely to be translocated from soil to plants compared to short chain, which are more water soluble [35]. The presence of PFAS in *Amaranthus viridis*, an edible plant, is of great concern, since edible parts of plants enriched with PFAS are considered and should be classified as substances of very high concern (SVHC) [36].

This study observed that long-chain PFAS levels were higher than the short-chain PFAS levels in water and sediment samples. Nairobi River was observed to contain a lot of bio solids and suspended material. The elevated level of long-chain PFAS could be ascribed to the fact that they are more bio-accumulative in the various environmental matrices [37] and are less susceptible to translocation to plants [35]. This observation contradicts other studies [17,38] which showed that short-chain PFAS have higher concentration levels in water and sediments attributed to partial breakdown of long-chain PFAS, and due to regulations in manufacturing that favor short-chain PFAS [39,40]. The higher levels of long-chain PFAS in sediments and water in the Nairobi River is of great concern since it suggests human exposure to these compounds.

### 3.1. Sediment/Water PFAS Distribution Coefficient

The distribution coefficient (log *Kd* values) is used as an indicator to assess the fate of organic contaminants [41]. The PFAS sediment/water distribution coefficient, *Kd*, in the Nairobi River was calculated using Equation (1) [42,43].
(1)Kd=Cs/Cw × 1000
where *Cs* (ng g^−1^) is the PFAS level in sediments, *Cw* (ng L^−1^) is the PFAS level in water. Theoretically, when estimated values for log *Kd* are above 1 the sediment is considered to be a depot for pollutants in aquatic systems [44]. Table 2 shows the distribution coefficients of selected PFAS.

Most compounds had a mean distribution coefficient value above 3 as shown in Table 2. The mean log *Kd* values ranged between 2.5 (PFuDA) and 4.9 (PFOS). The log *Kd* values of PFSAs were higher than the log *Kd* values of PFCAs; for instance, PFOS had the highest average log *Kd* value of 4.8 followed by PFBS (4.4) while PFOA had a mean log *Kd* value of 3.9 and PFBA had a log *Kd* value of 3.2. This shows that the sorption of sulfonates is stronger than the analogue carboxylic acids [45]. The high log *Kd* values obtained here indicate that sediments act as a reservoir of PFAS and hence a major PFAS source in the food web. The log *Kd* values of a given compound varied from place to place because different sediment properties and water conditions affect PFAS partition/portioning in the environment [46].

### 3.2. Comparison of PFAS Levels in Nairobi River to Other Studies in Africa

The results of PFAS levels in Nairobi River water obtained in this study can be compared to the levels of PFAS in surface water obtained in other studies in Africa.

The concentration levels of PFOA and PFOS obtained in this study are lower than the levels obtained in Kenya in previous studies such as in River Sossian and Lake Victoria Gulf as shown in Table 3. The lower levels of PFOA and PFOS in the Nairobi River could be attributed to the global campaign to reduce the use of long-chain PFAS.

The concentrations of PFOA and PFOS in Nairobi River water are higher compared to the concentrations observed in Ethiopia but are similar to the levels observed in Uganda. The levels of PFOA and PFOS in the Nairobi River are at least 10 times lower than levels reported in Nigeria and South Africa. Ethiopia is less developed compared to Kenya, while Nigeria and South Africa are more developed compared to Kenya. The use of products containing PFAS is expected to be higher in more developed countries [9]; thus, the observed trends could indicate less use of products containing PFAS in Ethiopia than in Kenya while products containing PFAS are widely used in Nigeria and South Africa.

## 4. Conclusions

This study presents new knowledge on occurrence and distribution of PFAS in water, sediments, and plants in the Nairobi River basin. Perfluoroalkyl substances are widely distributed within the environmental matrices in the Nairobi River basin, a total of 30 PFAS were detected in the river water, 28 PFAS congeners were found in sediments and plants. There were elevated levels of PFAS in sampling points near cottage industries. Long-chain PFAS were predominant in water and sediments while short-chain PFAS were predominant in plant samples. The concentration range of PFOA and PFOS in the water was 0.16–3.0 and 0.004–1.4 ng L^−1^, respectively. The log distribution coefficient values obtained indicated a preference of PFOS for sediment matrices (*Kd* value 4.9). The study offers an insight into the present state of PFAS pollution in the river, providing crucial information to the public and the government on the quality of water with respect to PFAS. The data presented can initiate the development of guidelines and management strategies.

## Figures and Tables

**Figure 1 ijerph-19-08980-f001:**
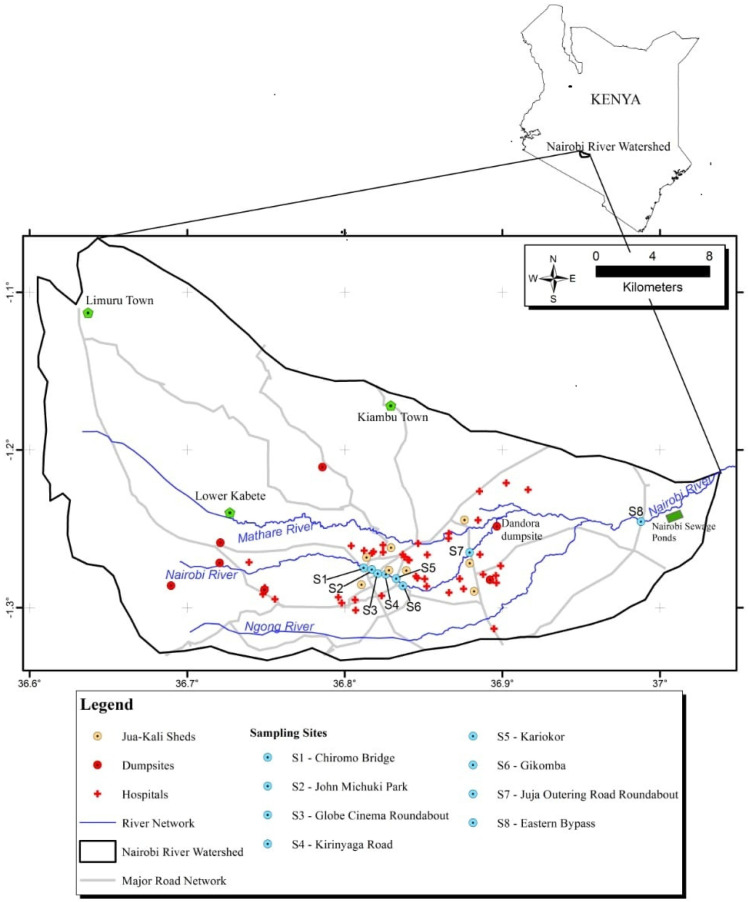
Study area and selected sampling points.

**Figure 2 ijerph-19-08980-f002:**
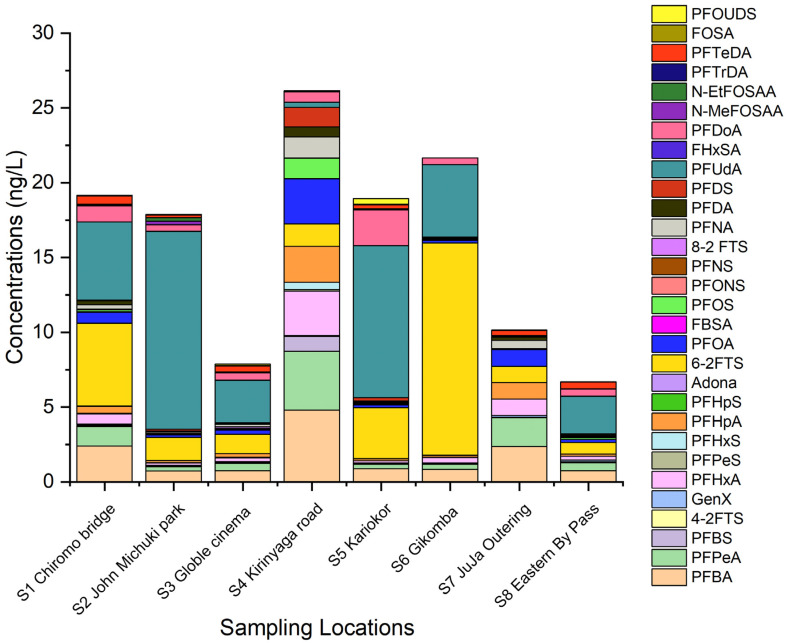
PFAS spatial distribution in Nairobi River water.

**Figure 3 ijerph-19-08980-f003:**
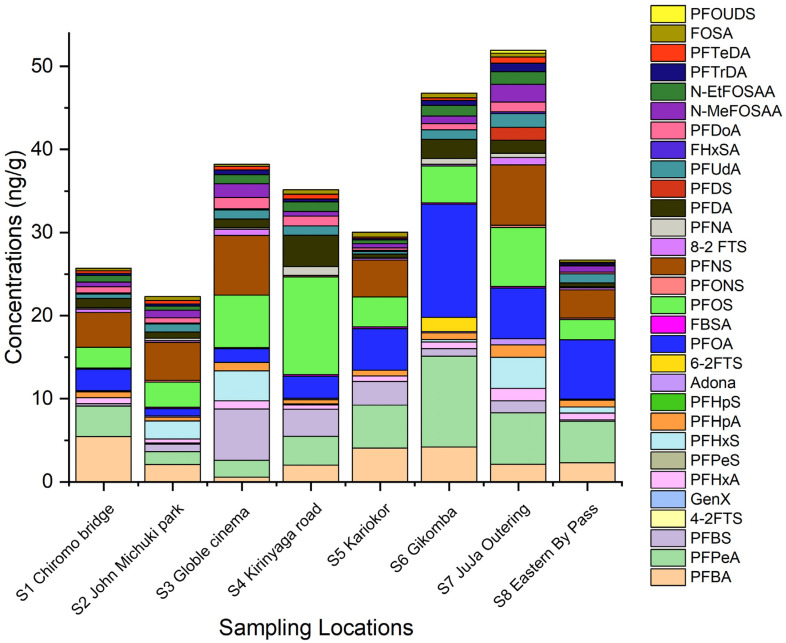
PFAS spatial distribution in sediments from the Nairobi River Basin.

**Figure 4 ijerph-19-08980-f004:**
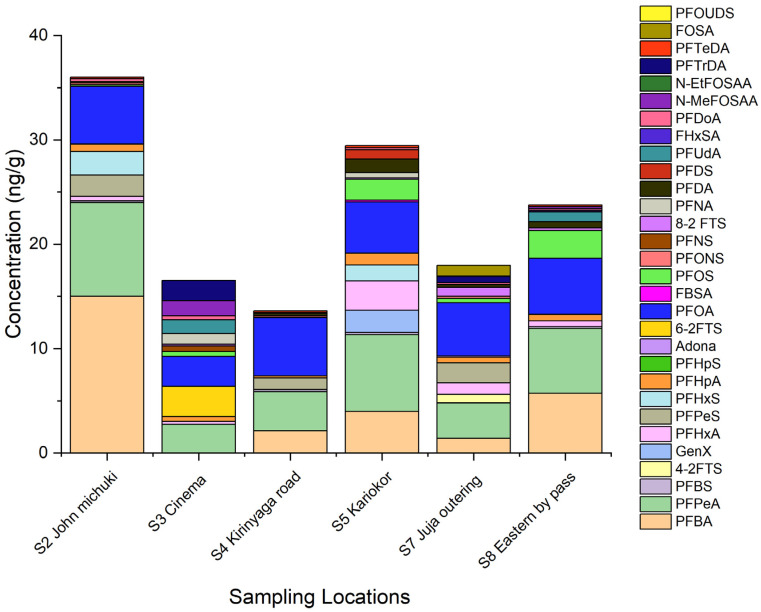
PFAS spatial distribution in plant samples from the Nairobi River Basin.

**Figure 5 ijerph-19-08980-f005:**
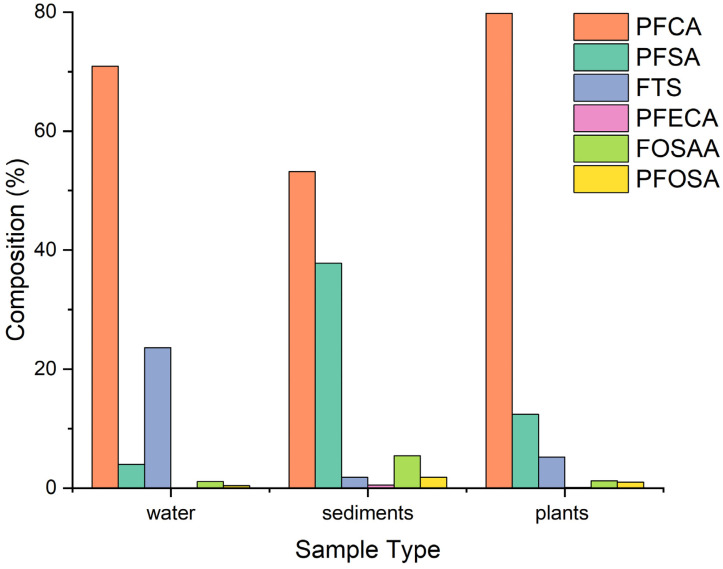
Percentage composition of different classes of PFAS in water, sediments, and plants.

**Figure 6 ijerph-19-08980-f006:**
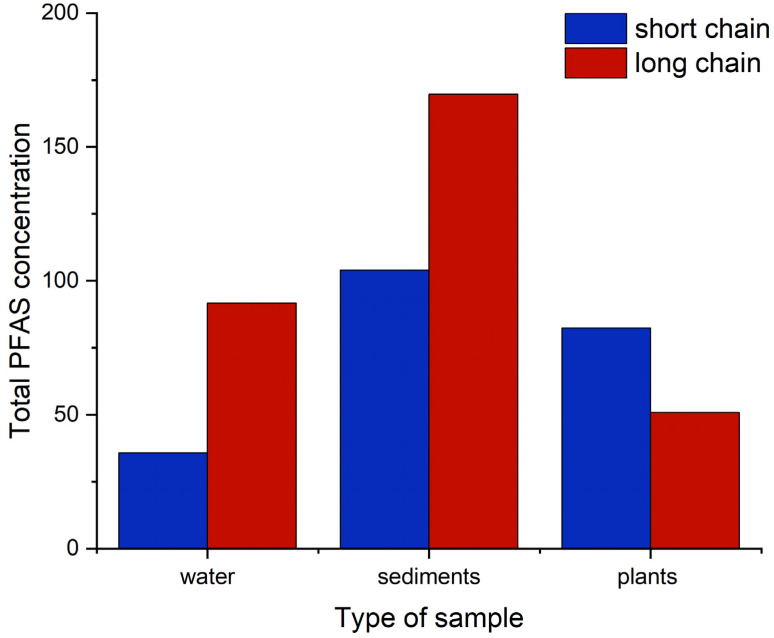
Classification of PFAS according to chain length in water, sediments, and plants. Concentrations are in ng g^−1^ for sediments and plants and ng L^−1^ for water.

**Table 1 ijerph-19-08980-t001:** Sampling point, geographic coordinate system, and description of the major anthropogenic activities.

Sampling Point	Designation	GIS Co-Ordinates	Description of Major Activities
Chiromo Bridge/Museum Hill roundabout	S1	−1.274697, 36.811925	Upstream with learning institutions, offices, and motor vehicles traffic
John Michuki Park	S2	−1.275753, 36.817161	Upstream small businesses, shops, cottage industries such as welding, car repair/garage
Globe Cinema roundabout	S3	−1.278397, 36.820902	Middle stream with motor vehicles traffic, cottage industries such as welding, car repair/garage
Kirinyaga road	S4	−1.279823, 36.825661	Middle stream with numerous cottage industries such as paints, panel beating, car repair/garage, dyeing industries
Kariokor	S5	−1.281559, 36.832641	Middle stream, small businesses, plastic and rubber burning with numerous cottage industries such as paints, panel beating, car repair/garage, dyeing industries, textile and food vending, dumpsites, wastewater inlet streams
Gikomba	S6	−1.286105, 36.836928	Middle stream with numerous cottage industries such as paints, panel beating, car repair/garage, dyeing industries, textile and food vending, wastewater inlet streams
Juja Outering road	S7	−1.264953, 36.879202	Downstream with leaning institutions, hospitals, dumpsites, offices and motor vehicles traffic, residential area
Eastern bypass	S8	−1.245381, 36.988022	Downstream with leaning institutions, residential area

**Table 2 ijerph-19-08980-t002:** The distribution coefficient values of selected PFAS in water and sediments.

Sampling Site	PFBA	PFPeA	PFHxA	PFHpA	PFOA	PFNA	PFDA	PFUDA	PFDoA	PFTeDA	PFBS	PFOS
S1	3.4	3.5	3	3.2	3.5	2.7	3.6	2	2.8	2.8	3.8	4.3
S2	3.5	3.7	3.4	3.5	3.8	3.6	4	1.8	3.2	3.4	4.7	4.9
S3	2.9	3.6	3.6	3.6	3.8	3.1	4.1	2.6	3.4	3.1	5.1	4.9
S4	2.6	2.9	2.3	2.4	3	3.9	3.7	3.4	3.2	-	3.5	3.9
S5	3.7	4.3	3.7	3.7	4.4	2.9	4	1.5	2.1	2.8	5.4	4.9
S6	3.7	4.5	3.4	3.8	4.9	4.1	4.7	1.4	3.2	-	4.4	5
S7	3	3.5	3.1	3.1	3.7	3	3.9	4.5	4.6	3.3	4.9	6.2
S8	3.5	4	3.5	3.7	4.6	3.3	3.7	2.6	2.6	2.4	3.5	4.3
Mean	3.3	3.8	3.3	3.4	4	3.1	4	2.5	3.2	3.3	3.9	4.9

Key: Log *Kd* value not calculated. S1 to S8 as designated in Table 1.

**Table 3 ijerph-19-08980-t003:** Comparison of the levels of PFAS in surface water in this study and other studies in Africa.

Country	Study Area	PFOA Range	PFOS Range	Most Common PFAS	Notable PFAS Source	Reference
Kenya	Nairobi River	0.16–3.0	0.004–1.4	PFUdA	Cottage industry	This study
Kenya	River Sosiani Eldoret	1.6–8.8	-	-	Industrial and domestic wastewater	[11]
Kenya	Lake Victoria Gulf	0.4–11.7	0.4–2.53	-	Industrial and urban wastewater	[13]
Ethiopia	Lake Tana	<0.28–0.69	0.055–0.22	PFBA, PFHxA	Wastewater from Bahir Dar	[47]
Uganda	Lake Victoria and lake Nakivoko	2.4	1.6	PFBS	Industrial and domestic discharge	[48]
Nigeria	A number of rivers within the country	0.8–2.8	3.9–10.1	PFOS	Industrial, domestic, and agricultural wastewater	[49]
South Africa	Vaal River	0.6–4.6	<0 LOD–35.7	PFOS	Mining industry and wastewater treatment plants	[50]
South Africa	Diep Western Cape	1.7–314	<LOD–183	PFOA	Urban, industrial, and agricultural discharges	[41]
Pore water Nigeria	A number of rivers within the country	4.7–11.1 (1.7)	10.9–20.4	PFOS	Domestic and industrial discharges	[51]

## Data Availability

Additional data is available as Appendix A.

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
