# Peer review of "Occurrence and Distribution of Per- and Polyfluoroalkyl Substances from Multi-Industry Sources to Water, Sediments and Plants along Nairobi River Basin, Kenya"

_ijerph, 2022, doi:10.3390/ijerph19158980_

Round 1

Reviewer 1 Report

How to identify the upstream, middle stream and downstream should be given.

Plant samples should be given more details which will told the audience how the PFASs were adsorbed and its origin.

The format of the references should be revised based on the guideline of this journal.

Author Response

Please see the attachment: Response to reviewer 1 comments 

Reviewer 2 Report

This MS reports the assessment of PFAS in environmental matrices along the Nairobi River, Kenya. The topic is relevant and interesting to the readers. 

Abstract:

Line 20: please provide the abbreviations in full form in the first place (i.e. PFOS). Please check and revise similar other cases as well. 

Introduction: 

Lines 32-36: Pelase provides the reference (source). This is one potential citation source. Stockholm Convention - Home page (pops.int)

The introduction needs to be improved a lot. Pelase add the environmental concentrations of PFAS, PFOS, etc information.  Also the sources of these pollutants.  

Methods: 

Please move the description of the study area section into the methods section. 

Conclusions: This needs to be improved a lot. The conclusion should be specific and provide the key findings of the study (including numerical concentrations etc. whenever relevant).  

Author Response

Please see the attachment. Response to reviewer 2 comments.
